

# Association of thyroid hormones with the severity of chronic kidney disease: a cross-sectional observational study at Tabuk, Saudi Arabia

Basmah Awwaadh[1], Amal Hussain Mohammed[1], Basmah F. Alharbi[2], Abdulmohsen Alruwetei[1], Tarique Sarwar[1], Hajed Obaid Alharbi[1] and Fahad Alhumaydhi[1]

[1] Department of Medical Laboratories, College of Applied Medical Sciences, Qassim University, Buraydah, Saudi Arabia
[2] Department of Basic Health Sciences, College of Applied Medical Sciences, Qassim University, Buraydah, Qassim Region, Saudi Arabia

## ABSTRACT

**Background**. The interplay between chronic kidney disease (CKD) and thyroid dysfunction is becoming more evident in the biomedical community. However, the intricacies of their relationship warrant deeper investigation to understand the clinical implications fully.

**Objective**. This study aims to systematically evaluate the correlation between thyroid hormone levels, including thyroid-stimulating hormone (TSH), triiodothyronine (T3), and thyroxine (T4), and markers of renal disease severity. These markers include serum creatinine, urea, and parathyroid hormone (PTH) levels in individuals diagnosed with CK).

**Methods**. We conducted a cross-sectional observational study involving a cohort of 86 participants with CKD recruited from the renal clinic at King Fahad Hospital in Tabuk. Biochemical parameters, encompassing plasma electrolytes and thyroid hormone concentrations, were quantitatively assessed. These measurements were performed with the aid of a Roche Cobas E411 analyzer. The Pearson correlation coefficient was employed to delineate the strength and direction of the associations between the thyroid function markers and renal disease indicators.

**Results**. The statistical analysis highlighted a generally weak correlation between the concentrations of thyroid hormones and the indicators of renal disease severity, with Pearson correlation coefficients between $-0.319$ and $0.815$. Critically, no significant correlation was found between creatinine and thyroid hormones (TSH, T3, T4), nor was any substantial correlation between urea and thyroid hormones. Conversely, a robust positive correlation was noted between the levels of parathyroid hormone and serum creatinine ($r = 0.718$, $p < 0.001$).

**Conclusion**. The data suggests that thyroid hormone levels have a minimal correlation with the severity of renal disease markers. In contrast, the pronounced correlation between PTH and creatinine underscores the importance of considering PTH as a significant factor in managing and therapeutic intervention of CKD complications.

Corresponding author
Fahad Alhumaydhi,
f.alhumaydhi@qu.edu.sa

These initial findings catalyze further research to thoroughly investigate the pathophysiological relationships and potential therapeutic targets concerning thyroid dysfunction in patients with renal impairment.

## INTRODUCTION

Among non-communicable diseases, the trajectory of mortality associated with chronic kidney disease (CKD) has been notably grim, escalating by 31.7% over the past decade—as highlighted by recent longitudinal studies—and is now recognized as a leading and rapidly growing cause of death, alongside ailments such as dementia and diabetes (*Wang et al., 2023*). Thyroid hormones, specifically triiodothyronine (T3) and thyroxine (T4)—with the latter also referred to as tetraiodothyronine due to its four iodine atoms—are imperative for renal ontogenesis and function from embryogenesis through adulthood. Current research acknowledges a substantial incidence of thyroid dysfunctions among patients with renal pathologies, particularly those receiving hemodialysis (*Li et al., 2023*; *Nwosu et al., 2022*). Despite the clear epidemiological association, the pathophysiological underpinnings linking the impairment of thyroid function to renal disease remain elusive. CKD's impact on systemic mineral balances—especially iodine, an element critical for thyroid hormone synthesis—provides a partial explanation. Moreover, it has been identified that the dysregulated homeostasis of iodine in CKD, leading to its accumulation, could precipitate both hypo- and hyperthyroidism through mechanisms such as the Wolff–Chaikoff effect and the Jod-Basedow phenomenon, respectively (*Peters et al., 2021*).

Recent investigations indicate that thyroid-stimulating hormone (TSH) levels exhibit a negligible rise in dialysis patients with concomitantly reduced T4, although this correlation is not pronounced (*Soylu et al., 2023*). However, a body of literature posits TSH as a robust biomarker for identifying CKD (*Schairer et al., 2020*; *Chen et al., 2020*; *Narasaki, Sohn & Rhee, 2021*; *Kumar, Saxena & Singh, 2023*). The intersection of CKD with disturbances in parathyroid hormone (PTH) signaling emphasizes the derangement of calcium and phosphate homeostasis, often culminating in secondary hyperparathyroidism. This complicates CKD management significantly, necessitating attentive monitoring and strategic intervention to attenuate bone and mineral metabolism disorders (*Yuasa et al., 2020*; *Li et al., 2020*; *Kwong et al., 2021*; *Tapper et al., 2021*; *Yang et al., 2022*).

PTH plays an essential role in regulating phosphorus and calcium metabolism, with its production being profoundly influenced by renal function. CKD impairs the kidneys' capacity for phosphorus filtration and the conversion of inactive to active vitamin D forms, both processes intricately tied to PTH secretion (*Sharma et al., 2022*; *Liu et al., 2023*). Reports show a heightened predisposition to thyroid pathologies, particularly hypothyroidism, in patients with varying renal disease etiologies, although prevalence rates
may display geographical disparities (*Matsuoka-Uchiyama et al., 2022*; *Sinjari & Ibrahim, 2022*; *Naguib & Elkemary, 2023*).

While the confluence of PTH, renal function, and CKD is acknowledged, the precise mechanisms driving their interaction require further exploration. The complex relationship involving PTH, phosphorus metabolism, and vitamin D in CKD is incompletely understood, particularly the signaling pathways and molecular entities that dictate PTH secretion in response to shifts in phosphorus handling and vitamin D status through different stages of CKD (*Kritmetapak & Pongchaiyakul, 2019*; *Yang et al., 2011*). The literature also falls short in thoroughly examining the bidirectional influence where CKD progression may exacerbate the derangement of calcium and phosphorus metabolism, leading to a complex interplay of cause and effect with PTH dysregulation (*Ammirati, 2020*; *Bello et al., 2017*). Understanding the clinical relationship between CKD and thyroid dysfunction is vital as thyroid issues, especially hypothyroidism, are more prevalent in CKD patients, affecting up to 20% (*Chonchol et al., 2008*). Thyroid hormones play a key role in metabolism and cardiovascular health, and their dysfunction can worsen cardiovascular complications, impacting CKD progression (*Rhee et al., 2015*). Proper management of thyroid dysfunction aids in addressing CKD-related complications like dyslipidemia, anemia, and bone disease, thereby improving patient outcomes. Additionally, overlapping symptoms between thyroid dysfunction and CKD can complicate diagnoses, making understanding this relationship crucial for accurate diagnosis and treatment (*Webster et al., 2017*). These factors highlight the need for exploring this intersection to optimize treatment strategies This research intends to close the existing knowledge chasms by analyzing the reciprocal interactions among PTH, renal physiology, and CKD. This study carefully selected thyroid hormones (TSH, T3, T4) and renal function markers (creatinine, urea) due to their roles in assessing thyroid and kidney health, crucial for understanding thyroid dysfunction's impact on CKD. The chosen cohort size ensures enough statistical power to reliably detect associations, enhancing the study's validity in exploring interactions between thyroid function and renal health in CKD patients.

## METHODOLOGY

### Study design

The investigation utilized a prospective cross-sectional observational framework. It included a cohort of 86 individuals attending the renal clinic at the King Fahad Specialist Hospital in Tabuk—an experienced nephrologist supervised data collection. The Tabuk Health Ethics Committee (H-07-TU-077) conferred explicit approval for the study. We received written informed consent from the participants in our study. The panel of tests performed comprised a comprehensive suite of electrolytes (including serum creatinine, blood urea nitrogen, potassium (K+), sodium (Na+)), along with a full thyroid panel (including parathyroid hormone (PTH), thyroid-stimulating hormone (TSH), free thyroxine (FT4), and free triiodothyronine (FT3)).

 

## Venous blood sampling protocol

Each subject provided approximately 10 mL of venal blood, which was apportioned into two containers: five mL was placed into EDTA tubes to facilitate electrolyte analysis, and the other five mL was aliquoted to serum separator tubes for hormone level assessment. All samples were centrifugated at 3,000 revolutions per minute, persisting for 5 min, to facilitate serum and plasma separation.

## Geographic and temporal context

The research was conducted at King Fahad Specialist Hospital within the urban area of Tabuk City in the period extending from September to December 2023.

## Participant cohort and sampling method

The sample included 86 residents of Saudi Arabia exhibiting pre-dialysis kidney function compromise, as evidenced by elevated blood urea, serum creatinine, and BUN levels. The diagnosis of CKD was based on these deranged kidney function tests. No USG evidence was collected, and only known cases of CKD were considered. individuals individual Sample size estimation was executed utilizing the Raosoft calculator, which integrates multiple parameters: the target population size, hypothesized prevalence or response distribution, margin of error, and desired confidence level. Raosoft computed a minimum sample size of 80 to achieve a 95% confidence level under recommendations delineated in the *McCrum-Gardner (2010)* discourse. The number of participants was adjusted to 86 to harmonize with comparable precedent studies of a larger scale. Anthropometric measurements refer to the assessment of various physical dimensions of the human body. In our study, we measured height, weight, body mass index (BMI), waist-to-hip ratio, *etc*. To obtain these measurements, we utilized tools such as measuring tapes, calipers, and stadiometers, ensuring precise and reliable data collection. Physiological measurements, on the other hand, focus on evaluating body functions. In our analysis, this included monitoring heart rate, blood pressure, respiratory rate, and body temperature. For these physiological assessments, we employed sphygmomanometers to measure blood pressure, thermometers to record body temperature, and biometric monitors to track heart rate. The use of these devices allowed for accurate and consistent evaluation of the participants' physiological status.

## Inclusion criteria

Selection involved adult candidates presenting renal pathologies as evinced by heightened serum creatinine and blood urea levels. Exclusion criteria barred ostensibly healthy candidates, individuals undergoing dialysis, kidney injury, and patients with concurrent systemic diseases.

## Data acquisition methods

Information was appropriated by deploying a structured interrogation instrument to compile sociodemographic variables. Moreover, blood specimen retrieval was performed for subsequent biochemical scrutiny, allowing for the assessment of thyroid functionality and renal efficiency.

### Analytic procedures

We analyzed the dataset by deploying SPSS software. The analytic techniques encompassed correlation and linear regression analyses, aiming to evaluate the dataset's interplay and potential predictive relationships. We computed Pearson correlation coefficients to evaluate the statistical dependence between thyroid endocrine parameters (TSH, FT4, FT3) and kidney function metrics (serum creatinine, blood urea nitrogen, K+, Na+). Regression analyses were next employed to ascertain the predictive potency of thyroid indicators concerning renal function markers. The objective was to discern statistically significant associations and delineate potential functional dependencies between thyroid hormone levels and indices of kidney pathology.

### Measurement techniques

Assays for T3, PTH, creatinine, TSH, and urea were executed using the Roche Cobas E411 analytical platform, conforming to the manufacturer's specifications for each test.

### Ethical compliance

The Tabuk Health Ethics Committee (H-07-TU-077) conferred explicit approval for the study. Before participation, individuals were adequately briefed on the study's scope and provided informed consent, with assurances of anonymity and the discretionary nature of their involvement.

### Primary outcome

The primary outcome of the study is to evaluate the correlation between thyroid hormone levels (TSH, T3) and the severity of chronic kidney disease (CKD) as indicated by renal function markers such as creatinine and urea levels.

### Secondary outcome

The secondary outcome includes assessing the relationship between parathyroid hormone (PTH) levels and renal function, as well as exploring the potential implications of thyroid hormone dysregulation on overall renal health and mineral metabolism in CKD patients This structured approach ensures a comprehensive understanding of the study's design and its intended outcomes. The study's methodology has been depicted as a flowchart in Fig. 1.

## RESULTS

### Relationships among serum creatinine levels, thyroid function parameters, and physiological measurements in patients

In exploring the association between serum creatinine levels and thyroid function, along with anthropometric and hemodynamic variables among the study participants, a series of Pearson correlation coefficients were calculated, as depicted in Table 1. The observed data reflect only tenuous associations between these variables. Serum creatinine levels showed a weak negative correlation with thyroid-stimulating hormone (TSH) ($r = -0.027$) and an even weaker inverse relationship with free thyroxine (T4) ($r = -0.111$), suggesting that as creatinine levels increase, there may be a slight, albeit statistically insignificant,

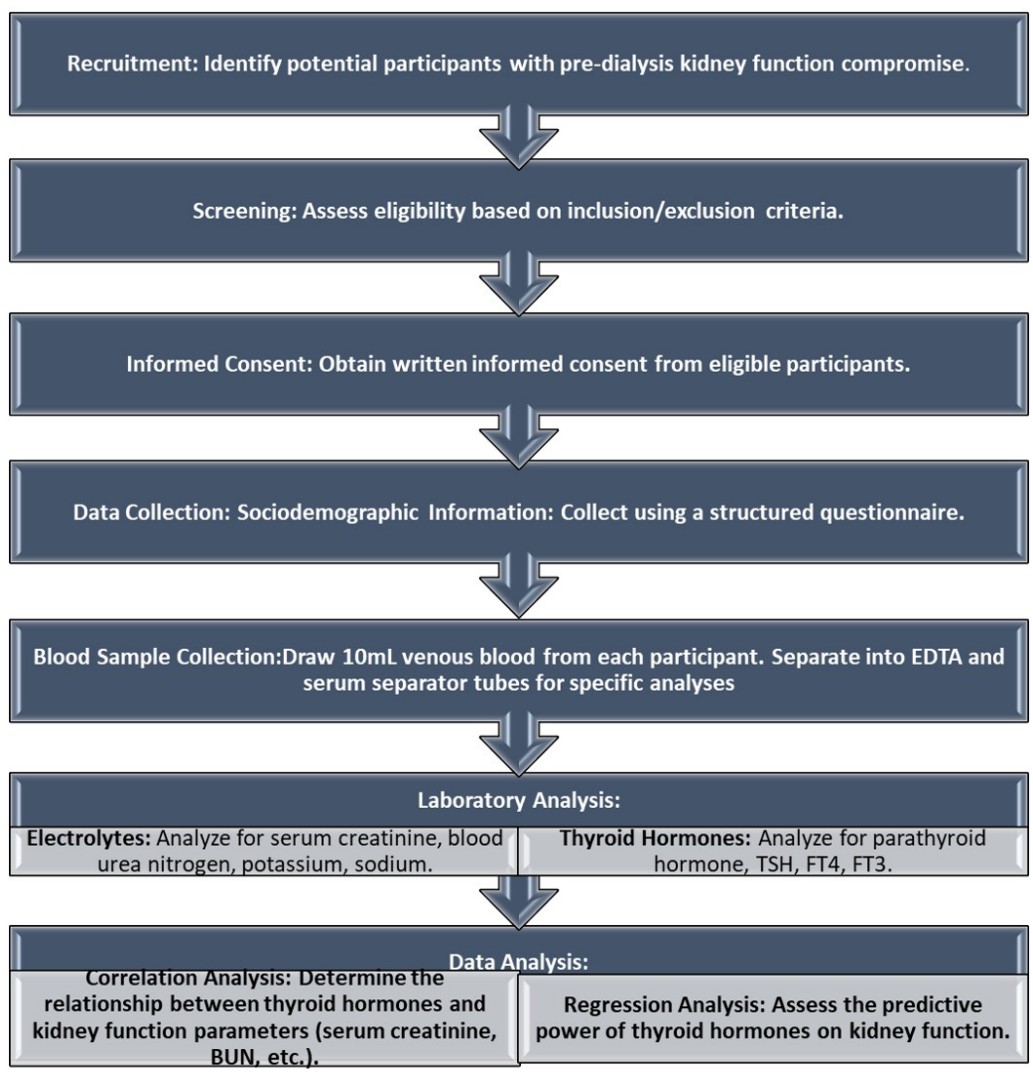

**Figure 1 Flowchart of the methodology.** Flowchart of the study's methodology.

decrease in TSH and T4 levels. Nonetheless, the association with triiodothyronine (T3) was in the positive direction ($r = 0.128$), indicating a marginal increase in T3 levels with rising creatinine; however, the clinical relevance of this finding is likely negligible given the lack of statistical significance. Regarding the anthropometric and hemodynamic measures, serum creatinine exhibited a very slight positive correlation with patient weight ($r = 0.030$) and a similarly minimal positive correlation with systolic blood pressure (SBP) ($r = 0.098$). These relationships suggest a practically inconsequential increase in weight and SBP with higher serum creatinine levels. Collectively, these data reflect the complexity of the interplay between renal function and thyroid hormones, as well as the relationship between renal function and body mass and systemic blood pressure regulation. The absence of statistically significant $p$-values, which ranged from 0.206 for the creatinine-T3 correlation to 0.768
**Table 1 Correlation of serum creatinine with thyroid function parameters and patient physiological measurements.**

|  |  | TSH | T3 | T4 | wt. | Bp |
|---|---|---|---|---|---|---|
| Creatinine | Pearson correlation | −.027 | .128 | −.111 | .030 | .098 |
|  | *P*-value | .794 | .206 | .276 | .768 | .338 |

for the creatinine-weight correlation, implies that serum creatinine concentrations are not reliable predictors of thyroid hormone levels, patient body weight, or SBP within this study cohort.

## Association between blood urea nitrogen levels, thyroid function parameters, and physiological measurements in patients

A comprehensive assessment was conducted to ascertain the interconnections between blood urea nitrogen (BUN) concentrations, various thyroid function tests, and the anthropometric and hemodynamic measures of patient weight and systolic blood pressure (SBP). As documented in Table 2, the Pearson correlation analysis was the statistical strategy employed to illuminate these relationships. Strikingly, the correlation coefficients manifest meager associations across all parameters. For thyroid-stimulating hormone (TSH), there was a nearly negligible inverse correlation with BUN ($r = -0.004$), implying no predictive or associative value between BUN and secretions of TSH. Analogously, the correlation with triiodothyronine (T3) was marginally positive ($r = 0.040$), while free thyroxine (T4) showcased a slightly inverse correlation ($r = -0.022$). These faint correlations with thyroid function tests may indicate that, within the study's context, BUN is not a substantial modifier or indicative marker of thyroid hormone fluctuations, contrary to situations in acute illness or more severe states of kidney dysfunction where effects on thyroid physiology are observed. Further extending this analysis to physical health determinants, the BUN correlation with patient weight surfaced as slightly negative ($r = -0.024$), hinting at an inverse relationship between urea concentrations and body mass. A negligible inverse correlation existed between BUN and SBP ($r = -0.018$). Even though these indicators are critical in evaluating patient health, particularly within nephrology and cardiovascular risk stratification, the data does not indicate a substantial relationship between urea levels and these anthropometric or hemodynamic parameters within this sample. None of the correlations identified were statistically significant, with p-values ranging from 0.694 for the BUN-T3 correlation to 0.967 for the BUN-TSH correlation. This lack of statistical significance reinforces the interpretation that variations in BUN levels do not reliably associate with the thyroid function measures or this cohort's chosen physical health parameters.

## Relationships between thyroid function parameters and patient physiological metrics

In the examination of thyroid regulatory dynamics and their implication on metabolic and cardiovascular parameters, Table 3 and Fig. 2 offer a visual and quantitative summary of the relationships between thyroid functions—specifically thyroid-stimulating hormone

**Table 2  Correlation of blood urea nitrogen with thyroid function parameters and patient physiological measurements.**

|  |  | TSH | T3 | T4 | wt. | Bp |
|---|---|---|---|---|---|---|
| Bun | Pearson correlation | −.004 | .040 | −.022 | −.024 | −.018 |
|  | *P*-value | .967 | .694 | .829 | .814 | .858 |

**Table 3  Correlations between thyroid function parameters and patient physiological measurements.**

|  | T3 | T4 | wt. | Bp |
|---|---|---|---|---|
| TSH | −.319 | .815 | .177 | .239 |
| T3 | – | .048 | .045 | .045 |
| T4 | – | – | −.030 | .043 |
| Wt. | – | – | – | −.061 |

(TSH), triiodothyronine (T3), and thyroxine (T4)—and variables of patient weight and systolic blood pressure (SBP). Remarkably, the Pearson correlation analysis has elucidated distinct patterns of association. The strength of the inverse relationships between TSH and the thyroid hormones T3 ($r = -0.319$, $p = 0.001$) and T4 ($r = -0.815$, $p < 0.001$) is pronounced. The magnitude of these negative correlations implies that higher TSH levels are systematically associated with lower levels of T3 and T4, a scenario often indicative of primary hypothyroidism, wherein the decreased thyroid function results in compensatory elevations of TSH. The particularly strong inverse relationship with T4 might also suggest a level of thyroid gland unresponsiveness or potential regulatory resistance. In contrast, TSH shows a moderate yet statistically significant correlation with patient weight ($r = 0.177$) and SBP ($r = 0.239$, $p = 0.019$).

## Influence of creatinine on blood parathyroid hormone (PTH) levels

Our examination of the indicative power of renal function markers to act as predictors for disturbances in the parathyroid hormone (PTH) secretion, which is pivotal in calcium and phosphate homeostasis, utilized regression analyses as detailed in Tables 4 and 5. The predictive capacity of such markers is critical for anticipating the onset and progression of secondary hyperparathyroidism, a common complication in chronic kidney disease (CKD). Table 4 illustrates the regression analysis outcomes that quantify the extent to which serum creatinine levels forecast PTH concentrations. A highly significant correlation ($p < 0.001$) was found, and the serum creatinine level was seen to exert a considerable influence on PTH variability within the patient population. The Pearson correlation coefficient of 0.718 denotes a strong positive association, wherein increases in serum creatinine levels correspond to corresponding elevations in PTH. This is indicative of renal dysfunction as a key driver for the stimulatory increase in PTH secretion, a compensatory response often triggered by diminishing renal clearance leading to phosphate retention and hypocalcemia. Moreover, the coefficient of determination ($R^2 = 0.516$) implies that more than half (approximately 51.6%) of the variance observed in PTH levels could be

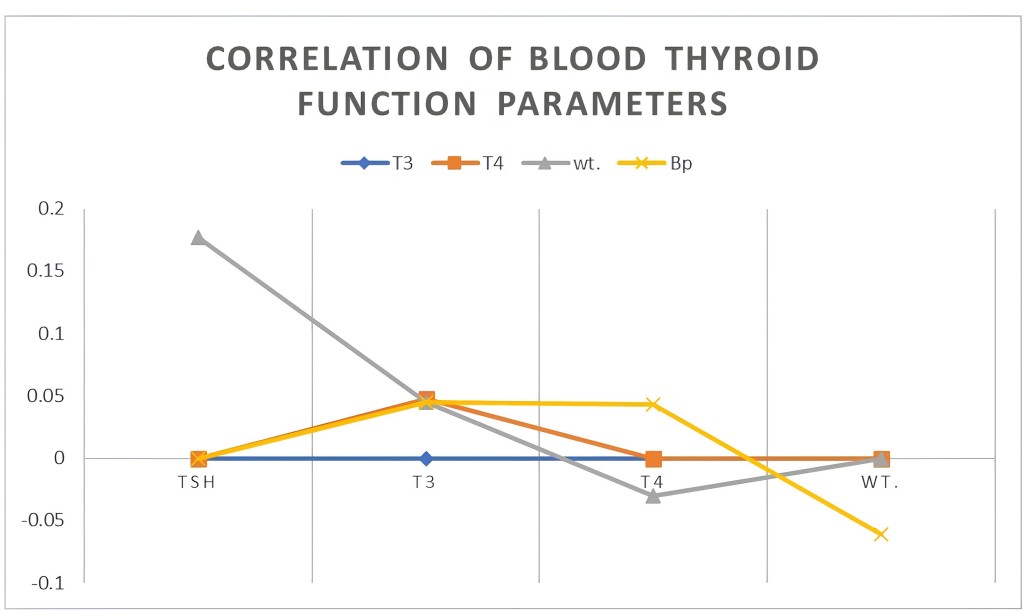

**Figure 2** Correlation of blood thyroid function parameters.

**Table 4 Impact of creatinine on blood parathyroid hormone (PTH).**

| | Independent variable creatinine | | | | Dependent variable PTH |
|---|---|---|---|---|---|
| Coefficient of determination ($R^2$) | Correlation (R) | *p*-value | (F) | B | |
| .516 | .718 | .001 | 88.517 | .127 | |

**Table 5 Impact of urea on blood parathyroid hormone (PTH).**

| | Independent Variable BUN | | | | Dependent variable: PTH |
|---|---|---|---|---|---|
| Coefficient of determination ($R^2$) | Correlation (R) | *p*-value | (F) | B | |
| .078 | .280 | .009 | 88.517 | 1.608 | |

attributed to the serum creatinine levels, an indication that renal impairment, as evidenced by elevated creatinine, is a substantial factor in PTH level alteration.

### Influence of urea on blood parathyroid hormone (PTH) levels

Table 5 encapsulates the findings from the regression analysis, shedding light on the connection between blood urea nitrogen (BUN) levels and parathyroid hormone (PTH) concentrations. Notably, the data unveils a statistically discernible association between these variables, elucidated by a *p*-value of 0.009. The Pearson correlation coefficient (r)

of 0.280 characterizes this association, signifying a positive, albeit more modest, linkage compared to the relationship observed with serum creatinine.

## DISCUSSION

The current study's findings demonstrate a statistically significant relationship between chronic kidney disease (CKD) markers, specifically creatinine and urea, and parathyroid hormone (PTH) levels. This association supports the established pathophysiology where elevated PTH, while critical for maintaining calcium homeostasis, negatively impacts bone health, potentially resulting in renal osteodystrophy (*Tawfik et al., 2022*). An increase in PTH typically serves as a compensatory response to the decline in blood calcium levels due to impaired renal function, particularly in CKD (*Al Fahdi et al., 2022*). Our investigation highlights a positive correlation between blood PTH, creatinine, and urea, aligning with findings by *Reque Santivañez et al. (2022)* and *Kim et al. (2023)*, who reported similar linkages in renal impairment contexts. Notably, as creatinine and urea levels rise, PTH also increases. This is critical as PTH acts as an important marker for CKD; excess PTH can contribute to kidney failure, indicating renal impairment when elevated (*Echterdiek et al., 2022*). The connection between elevated creatinine, urea, and increased PTH levels is linked with secondary hyperparathyroidism.

The kidneys are vital in regulating mineral and electrolyte balance, including phosphorus and calcium levels (*Zeng & Liao, 2022*). When kidney function is compromised, as evidenced by elevated creatinine and urea, phosphate excretion is often impaired, leading to phosphorus accumulation (*Ansari et al., 2023*). This condition triggers a decrease in calcium levels, prompting the parathyroid glands to release more PTH, which increases calcium levels by mobilizing calcium from bones and enhancing renal calcium reabsorption (*Alsulami et al., 2022*). The increase in PTH in response to elevated creatinine and urea levels compensates for the calcium-phosphorus imbalance due to impaired kidney function in CKD, where mineral metabolism disruptions can result in secondary hyperparathyroidism (*Geng et al., 2022*). However, there is a statistically insignificant relationship between creatinine, urea, patient weight, and blood pressure. These findings contrast with several previous studies, such as *Liu (2023)*, but show similarity to *Bamia et al. (2023)*.

Furthermore, exploring the interplay between PTH, patient weight, and blood pressure offers insightful information into PTH's systemic impacts on cardiovascular and metabolic parameters. Hyperthyroidism is linked to decreased vascular resistance and potential reductions in blood pressure, while hypothyroidism may elevate blood pressure and alter cardiac function (*Cheng et al., 2023*). Despite existing findings regarding PTH, creatinine, and urea, there remains much to uncover about the interactions between thyroid function, renal markers, and PTH in chronic kidney disease. Future research should address limitations and discrepancies observed and strive for a comprehensive understanding of these interactions to improve clinical management strategies. This could include longitudinal studies to monitor changes, expanded hormonal profiling, and the consideration of medications influencing thyroid and parathyroid functions. A high $R^2$ value suggests that various factors, such as genetic predispositions, CKD severity

and duration, dietary phosphorus intake, vitamin D status, and medication use, might contribute to the remaining variance. The significant correlation and $R^2$ value emphasize the critical pathophysiological link between renal function and PTH secretion. Given the kidneys' role in vitamin D activation and phosphate excretion, their dysfunction in CKD can trigger secondary hyperparathyroidism. As serum creatinine indicates glomerular filtration rate (GFR), these findings reinforce its utility in estimating renal function and its role in bone and mineral disorders. There is potential for predictive modeling using serum creatinine to assess secondary hyperparathyroidism risk and progression. Clinicians can leverage this relationship to optimize CKD management, aiming to reduce subsequent bone disease and vascular calcification risks. Future clinical trials and longitudinal studies should broaden biomarker profiles, exploring direct renal markers and the comprehensive hormone interplay in CKD-related bone and mineral disorders to refine models and individualize patient care.

The findings align with literature suggesting possible links between thyroid function, body weight regulation mechanisms, and blood pressure modulation. Beyond thyroid hormone synthesis, TSH may affect adipocytes and vascular resistance. However, observed correlations between T3, T4, weight, and systolic blood pressure (SBP) are minor and statistically insignificant (all $p$-values >0.05), indicating that T3 and T4 levels are non-predictive of these physiological measures' variations in this dataset. The effects of T3 and T4 on metabolic and cardiovascular systems are likely intricate, involving complex interactions with other endocrinological and metabolic factors not directly captured here. T3 is known to increase basal metabolic rate and cardiac output, impacting blood pressure, while T4's effects primarily occur after conversion to T3. The absence of a direct correlation suggests other regulatory mechanisms maintain hemodynamic stability and body weight despite thyroid hormone fluctuations. The coefficient of determination ($R^2 = 0.078$) implies about 7.8% of the PTH variability is due to changes in BUN, highlighting BUN's modest but distinct influence on PTH regulation. This underlines the consideration of BUN as a marker of renal function and nitrogen balance, alongside its impact on PTH dynamics. Both BUN and PTH are critical indicators of renal and mineral metabolic health, suggesting incorporating BUN-PTH dynamics into predictive models for mineral bone disorder progression in CKD will be valuable. Further research should explore the combined influence of BUN and metabolic parameters for deeper insights into their roles in the renal and mineral metabolic disorder context. This study effectively contributes to understanding the intricate relationships among thyroid hormones, renal markers, and PTH in CKD patients through its cross-sectional design. Despite its insights, further research with larger sample sizes, calculated statistical power, and longitudinal evaluation is required to confirm these findings and address acknowledged limitations. Addressing these considerations will enhance our understanding of the interactions between thyroid function, renal markers, and PTH in CKD, impacting clinical management strategies.

## CONCLUSION

This investigation has enhanced our understanding of the complex relationship between thyroid hormone levels and renal disease severity. By analyzing thyroid hormone levels

alongside renal dysfunction parameters, the study identifies significant associations with important clinical implications, emphasizing the need for close monitoring of thyroid function in individuals with renal disease, as thyroid dysfunction may affect disease progression and complications. Additionally, exploring the mechanistic pathways linking thyroid hormones and renal disease severity opens avenues for developing targeted interventions to improve patient outcomes. In CKD, the findings highlight the importance of thyroid function monitoring as part of a comprehensive management strategy, despite the limited direct impact of thyroid hormones on renal markers. The study also highlights the strong correlation between elevated parathyroid hormone (PTH) levels and creatinine, positioning PTH as a valuable marker for assessing renal impairment and guiding evaluations of CKD progression and therapeutic strategies. Ongoing research is essential to further explore the intricate relationships between thyroid function, renal markers, and PTH. Longitudinal studies involving larger cohorts could lead to targeted interventions aimed at enhancing mineral metabolism and renal health. By integrating findings related to both PTH and renal function, healthcare providers can improve clinical management strategies, ultimately leading to better outcomes for CKD patients. Collectively, these implications underscore the necessity for a holistic approach to CKD management, which considers both hormonal and renal factors to optimize patient care.

### Funding
This work was supported by the Deanship of Graduate Studies and Scientific Research at Qassim University (QU-APC-2024-9/1). The funders had no role in study design, data collection and analysis, decision to publish, or preparation of the manuscript.

### Grant Disclosures
The following grant information was disclosed by the authors:
Deanship of Scientific Research, Qassim University: QU-APC-2024-9/1.

### Competing Interests
The authors declare there are no competing interests.

### Author Contributions
- Basmah Awwaadh conceived and designed the experiments, performed the experiments, prepared figures and/or tables, and approved the final draft.
- Amal Hussain Mohammed performed the experiments, prepared figures and/or tables, and approved the final draft.
- Basmah F. Alharbi performed the experiments, prepared figures and/or tables, and approved the final draft.
- Abdulmohsen Alruwetei analyzed the data, authored or reviewed drafts of the article, and approved the final draft.
- Tarique Sarwar analyzed the data, authored or reviewed drafts of the article, and approved the final draft.

- Hajed Obaid Alharbi analyzed the data, authored or reviewed drafts of the article, and approved the final draft.
- Fahad Alhumaydhi conceived and designed the experiments, analyzed the data, authored or reviewed drafts of the article, and approved the final draft.

## Human Ethics

The following information was supplied relating to ethical approvals (i.e., approving body and any reference numbers):

The Tabuk Health Ethics Committee conferred explicit approval for the study (Ethical Application Reference: H-07-TU-077)

## Data Availability

The raw data is available in the Supplementary File.

## Supplemental Information

Supplemental information for this article can be found online at http://dx.doi.org/10.7717/peerj.18338#supplemental-information.

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
