# Peer review of "Association of thyroid hormones with the severity of chronic kidney disease: a cross-sectional observational study at Tabuk, Saudi Arabia"

_PeerJ, doi:10.7717/peerj.18338_

## Round 0.1 · original submission · Major Revisions

Dear authors,

Thank you for your submission. Apologies for the delay. It is consensual that your work requires very significative revisions. Please, refer to the reviewers' comments for details.

Reviewer 1 ·

Basic reporting

This article needs major revision, suggestions has been provided in the manuscript as sticky note.
All section like Title, introduction, method, result and discussion needs major changes.
needs professional english editing, as there are many grammerly errors.

Experimental design

Needs revision

Validity of the findings

Needs revision

Additional comments

NIL

Annotated reviews are not available for download in order to protect the identity of reviewers who chose to remain anonymous.

Reviewer 2 ·

Basic reporting

Correlation Between Thyroid Hormone Levels and Renal Disease Severity in Chronic Kidney Disease Patients" aims to investigate the relationship between thyroid hormone levels (TSH, T3, T4) and markers of renal disease severity (creatinine, urea, PTH) in patients with chronic kidney disease (CKD). The study employs a cross-sectional observational design involving 86 participants from King Fahad Hospital in Tabuk, Saudi Arabia. The research is structured well, with clear objectives outlined in the abstract and introduction sections. Key concepts are introduced effectively, providing a comprehensive background to contextualize the study's significance.

Experimental design

The study utilizes a prospective cross-sectional observational framework, which is appropriate for exploring associations between variables but limits the ability to establish causality. The cohort selection from a renal clinic and adherence to ethical guidelines are strengths of the study design. The methodology details the use of biochemical assays conducted on Roche Cobas E411 analyzers for thyroid hormones and renal markers, ensuring standardized measurements. However, the study lacks details on potential confounders or variables controlled for in the analysis, which could affect the interpretation of results.

Validity of the findings

The findings suggest a weak correlation between thyroid hormone levels and renal disease severity markers. Specifically, no significant correlations were found between creatinine/urea and thyroid hormones, contrasting with a strong positive correlation between PTH and creatinine. The use of Pearson correlation coefficients to quantify associations is appropriate, although the significance and clinical relevance of some correlations, such as T3 with creatinine, are questionable due to weak coefficients and lack of statistical significance. The strong correlation between PTH and creatinine highlights the study's contribution to understanding secondary hyperparathyroidism in CKD.

Additional comments

A few of the suggestions are listed.
Specify briefly why understanding the relationship between CKD and thyroid dysfunction is important from a clinical perspective. For instance, mention the prevalence or impact of thyroid dysfunction in CKD patients.
Consider rephrasing to explicitly state the primary aim of the study. For example, "This study aims to assess the correlation between thyroid hormone levels (TSH, T3, T4) and markers of renal disease severity in CKD patients."
Include a brief sentence explaining why these specific biochemical parameters and this cohort size were chosen. This adds context to the study design.
Instead of listing Pearson correlation coefficients, summarize the key findings in relation to the study objective. For instance, "Thyroid hormone levels showed weak correlations with markers of renal disease severity, while a significant positive correlation was found between PTH and serum creatinine.
Emphasize the implications of the findings for clinical practice and future research. For example, "While thyroid hormone levels appear to have limited impact on renal disease markers in CKD, the strong correlation between PTH and creatinine suggests potential implications for therapeutic strategies."
The study provides valuable insights into the complex interplay between thyroid function and renal disease markers in CKD patients. While the findings contribute to the existing literature, several methodological improvements and considerations for future research could strengthen the validity and applicability of the results.

---

## Round 0.2 · accepted · Accept

Dear authors, congratulations. Your manuscript is now accepted for publication in PeerJ. Many congratulations.

Reviewer 1 ·

Basic reporting

satisfied.

Experimental design

satisfied

Validity of the findings

satisfied

Additional comments

nil

Reviewer 2 ·

Basic reporting

The manuscript is written in clear, professional language and provides a thorough background, citing relevant studies linking CKD with thyroid hormone dysfunction, focusing on secondary hyperparathyroidism. References are appropriately cited and relevant.

Experimental design

The study addresses a significant issue, the relationship between thyroid hormone levels and CKD markers, filling an identified gap in understanding how thyroid function correlates with CKD progression. The study utilizes a cross-sectional observational design with a sample size of 86 CKD patients. The methods are well described . However, the exclusion criteria could be more robust by considering other comorbidities that might confound the findings. Pearson’s correlation coefficients were employed, with adequate attention given to determining the strength and direction of associations between thyroid function and CKD markers. While the correlations are generally weak, the methods chosen for data analysis are suitable for the scope of the study.

Validity of the findings

The data provided in the results section indicates weak correlations between thyroid hormones and renal function markers (TSH, T3, T4 with creatinine and urea). However, a significant relationship was found between parathyroid hormone (PTH) and creatinine, suggesting PTH may be a more relevant indicator in CKD management than thyroid hormones.

Additional comments

The study provides useful initial insights into the role of thyroid hormones and PTH in CKD. The weak correlation between thyroid hormones and renal markers may require further exploration with larger or more diverse patient cohorts.